# Decentralized Cooperative Stochastic Bandits

**David Martínez-Rubio**
Department of Computer Science
University of Oxford
Oxford, United Kingdom
david.martinez@cs.ox.ac.uk

**Varun Kanade**
Department of Computer Science
University of Oxford
Oxford, United Kingdom
varunk@cs.ox.ac.uk

**Patrick Rebeschini**
Department of Statistics
University of Oxford
Oxford, United Kingdom
patrick.rebeschini@stats.ox.ac.uk

## Abstract

We study a decentralized cooperative stochastic multi-armed bandit problem with $K$ arms on a network of $N$ agents. In our model, the reward distribution of each arm is the same for each agent and rewards are drawn independently across agents and time steps. In each round, each agent chooses an arm to play and subsequently sends a message to her neighbors. The goal is to minimize the overall regret of the entire network. We design a fully decentralized algorithm that uses an accelerated consensus procedure to compute (delayed) estimates of the average of rewards obtained by all the agents for each arm, and then uses an upper confidence bound (UCB) algorithm that accounts for the delay and error of the estimates. We analyze the regret of our algorithm and also provide a lower bound. The regret is bounded by the optimal centralized regret plus a natural and simple term depending on the spectral gap of the communication matrix. Our algorithm is simpler to analyze than those proposed in prior work and it achieves better regret bounds, while requiring less information about the underlying network. It also performs better empirically.

## 1 Introduction

The multi-armed bandit (MAB) problem is one of the most widely studied problems in online learning. In the most basic setting of this problem, an agent has to pull one among a finite set of arms (or actions), and she receives a reward that depends on the chosen action. This process is repeated over a finite time-horizon and the goal is to get a cumulative reward as close as possible to the reward she could have obtained by committing to the best fixed action (in hindsight). The agent only observes the rewards corresponding to the actions she chooses, i.e., the *bandit* setting as opposed to the full-information setting.

There are two main variants of the MAB problem—the stochastic and adversarial versions. In this work, our focus is on the former, where each action yields a reward that is drawn from a fixed unknown (but stationary) distribution. In the latter version, rewards may be chosen by an adversary who may be aware of the strategy employed by the agent, but does not observe the random choices made by the agent. Optimal algorithms have been developed for both the stochastic and the adversarial versions (cf. [9] for references). The MAB problem epitomizes the exploration-exploitation tradeoff that appears in most online learning settings: in order to maximize the cumulative reward, it is necessary to trade off between the exploration of the hitherto under-explored arms and the exploitation of the

seemingly best arm. Variants of the MAB problem are used in a wide variety of applications ranging from online advertising systems to clinical trials, queuing and scheduling.

In several applications, the "agent" solving the MAB problem may itself be a distributed system, e.g., [1, 10, 15, 33, 35, 36]. The reason for using decentralized computation may be an inherent restriction in some cases, or it could be a choice made to improve the total running time—using $N$ units allows $N$ arms to be pulled at each time step. When the agent is a distributed system, restrictions on communication in the system introduce additional tradeoffs between communication cost and regret. Apart from the one considered in this work, there are several formulations of decentralized or distributed MAB problems, some of which are discussed in the related work section below.

**Problem Formulation**. This work focuses on a *decentralized* stochastic MAB problem. We consider a network consisting of $N$ agents that play the *same* MAB problem *synchronously* for $T$ rounds, and the goal is to obtain regret close to that incurred by an optimal centralized algorithm running for $NT$ rounds ($NT$ is the total number of arm pulls made by the decentralized algorithm). At each time step, all agents simultaneously pull some arm and obtain a reward drawn from the distribution corresponding to the pulled arm. The rewards are drawn independently across the agents and the time steps. After the rewards have been received, the agents can send messages to their neighbors.

**Main Contributions**. We solve the decentralized MAB problem using a *gossip* algorithm.[1] Our algorithm incurs regret equal to the optimal regret in the centralized problem plus a term that depends on the spectral gap of the underlying communication graph and the number of agents (see Theorem 3.2 for a precise statement). At the end of each round, each agent sends $O(K)$ values to her neighbors. The amount of communication permitted can be reduced at the expense of incurring greater regret, capturing the communication-regret tradeoff. The algorithm needs to know the total number of agents in the network and an upper bound on the spectral gap of the communication matrix. We assume the former for clarity of exposition, but the number of nodes can be estimated, which is enough for our purposes (cf. Appendix F). The latter is widely made in the decentralized literature [7, 13, 14, 30].

The key contribution of our work is an algorithm for the decentralized setting that exhibits a natural and simple dependence on the spectral gap of the communication matrix. In particular, for our algorithm we have:

- A regret bound that is simpler to interpret, and asymptotically lower compared to other algorithms previously designed for the same setting. We use delayed estimators of the relevant information that is communicated in order to significantly reduce their variance.

- A graph-independent factor multiplying $\log T$ in the regret as opposed to previous works.

- Our algorithm is fully decentralized and can be implemented on an arbitrary network, unlike some of the other algorithms considered in the literature, which need to use extra global information. This is of interest for decentralization purposes but also from the point of view of the total computational complexity.

- We use accelerated communication, which reduces the regret dependence on the spectral gap, which is important for scalability purposes.

**Future work**. Decentralized algorithms of this kind are a first step towards solving problems on time-varying graphs or on networks prone to communication errors. We leave for future research an extension to time-varying graphs or graphs with random edge failures. Further future research can include a change in the model to allow asynchronous communication, making some assumptions on the nodes so they have comparable activation frequencies.

## 1.1 Related Work

**Distributed Algorithms**. The development of distributed algorithms for optimization and decision-making problems has been an active area of research, motivated in part by the recent development of large scale distributed systems that enable speeding up computations. In some cases, distributed computation is a necessary restriction that is part of the problem, as is the case in packet routing or sensor networks. *Gossip algorithms* are a commonly used framework in this area [7, 13, 14, 28, 30, 31]. In gossip algorithms, we have an iterative procedure with processing units at the nodes of a

graph and the communication pattern dictated by the edges of the graph. A common sub-problem in these applications is to have a value at each node that we want to average or synchronize across the network. In fact, most solutions reduce to approximate averaging or synchronization. This can be achieved using the following simple and effective method: make each node compute iteratively a weighted average of its own value and the ones communicated by its neighbors, ensuring that the final value at each node converges to the average of the initial values across the network. Formally, this communication can be represented as a multiplication by a matrix $P$ that respects the network structure and satisfies some conditions that guarantee fast averaging. The averaging can be accelerated by the use of Chebychev polynomials (see Lemma 3.1).

**Decentralized Bandits**. There are several works that study stochastic and nonstochastic distributed or decentralized multi-armed bandit problems, but the precise models vary considerably.

In the stochastic case, the work of Landgren et al. [24, 25] proposes three algorithms to solve the same problem that we consider in this paper: coop-UCB, coop-UCB2 and coop-UCL. The algorithm coop-UCB follows a variant of the natural approach to solve this problem that is discussed in Section 3. It needs to know more global information about the graph than just the number of nodes and the spectral gap: the algorithm uses a value per node that depends on the whole spectrum and the set of eigenvectors of the communication matrix. The algorithm coop-UCB2 is a modification of coop-UCB, in which the only information used about the graph is the number of nodes, but the regret is significantly greater. Finally, coop-UCL is a Bayesian algorithm that also incurs greater regret than coop-UCB. Our algorithm obtains lower asymptotic regret than all these algorithms while retaining the same computational complexity (cf. Remark 3.4).

Our work draws on techniques on gossip acceleration and stochastic bandits with delayed feedback. A number of works in the literature consider Chebyshev acceleration applied to gossip algorithms, e.g., [2, 30]. There are various works about learning with delayed feedback. The most relevant work to our problem is [19] which studies general online learning problems under delayed feedback. Our setting differs in that we not only deal with delayed rewards but with approximations of them.

Several other variants of distributed stochastic MAB problems have been proposed. Chakraborty et al. [12] consider the setting where at each time step, the agents can either broadcast the last obtained reward to the whole network or pull an arm. Korda et al. [22] study the setting where each agent can only send information to one other agent per round, but this can be any agent in the network (not necessarily a neighbor). Szörényi et al. [34] study the MAB problem in P2P random networks and analyze the regret based on delayed reward estimates. Some other works do not assume independence of the reward draws across the network. Liu and Zhao [26] and Kalathil et al. [20] consider a distributed MAB problem with collisions: if two players pull the same arm, the reward is split or no reward is obtained at all. Moreover in the latter work and a follow-up [27], the act of communicating increases the regret. Anandkumar et al. [1] also consider a model with collisions and agents have to learn from *action collisions* rather than by exchanging information. Shahrampour et al. [32] consider the setting where each agent plays a different MAB problem and the total regret is minimized in order to identify the best action when averaged across nodes. Nodes only send values to their neighbors but it is not a completely decentralized algorithm, since at each time step the arm played by all the nodes is given by the majority vote of the agents. Xu et al. [38] study a distributed MAB problem with global feedback, i.e., with no communication involved. Kar et al. [21] also consider a different distributed bandit model in which only one agent observes the rewards for the actions she plays, while the others observe nothing and have to rely on the information broadcast by the first agent.

The problem of identifying an $\varepsilon$-optimal arm using a distributed network has also been studied. Hillel et al. [16] provide matching upper and lower bounds in the case that the communication happens only once and when the graph topology is restricted to be the complete graph. They provide an algorithm that achieves a speed up of $N$ (the number agents) if $\log 1/\varepsilon$ communication steps are permitted.

In the adversarial version, the best possible regret bound in the centralized setting is still $\sqrt{KT}$ [3]. In the decentralized case, a trivial algorithm that has no communication incurs regret $N\sqrt{KT}$; and a lower bound of $N\sqrt{T}$ is known [11]; thus, only the dependence on $K$ can be improved. Awerbuch and Kleinberg [6] study a distributed adversarial MAB problem with some Byzantine users, i.e., users that do not follow the protocol or report fake observations as they wish. In the case in which there are no Byzantine users they obtain a regret of $O(T^{2/3}(N + K)\log N \log T)$. To the best of our knowledge, this is the first work that considers a decentralized adversarial MAB problem. They

allow $\log(N)$ communication rounds between decision steps so it differs with our model in terms of communication. Also in the adversarial case, Cesa-Bianchi et al. [11] studied an algorithm that achieves regret $N(\sqrt{K^{1/2}T \log K} + \sqrt{K} \log T)$ and prove some results that are graph-dependent. The model is the same as ours, but in addition to the rewards she obtained, each agent communicates to her neighbors all the values she received from her neighbors in the last $d$ rounds, that is potentially $O(Nd)$. Thus, the size of each message could be more than poly$(K)$ at a given round. They get the aforementioned regret bound by setting $d = \sqrt{K}$.

## 2 Model and Problem Formulation

We consider a multi-agent network with $N$ agents. The agents are represented by the nodes of an undirected and connected graph $G$ and each agent can only communicate to her neighbors. Agents play the same $K$-armed bandit problem for $T$ time steps, send some values to their neighbors after each play and receive the information sent by their respective neighbors to use it in the next time step if they so wish. If an agent plays arm $k$, she receives a reward drawn from a fixed distribution with mean $\mu_k$ that is independent of the agent. The draw is independent of actions taken at previous time steps and of actions played by other agents. We assume that rewards come from distributions that are subgaussian with variance proxy $\sigma^2$.

Assume without loss of generality that $\mu_1 \geq \mu_2 \geq \cdots \geq \mu_K$, and let the suboptimality gap be defined as $\Delta_k := \mu_1 - \mu_k$ for any action $k$. Let $I_{t,i}$ be the random variable that represents the action played by agent $i$ at time $t$. Let $n_{t,i}^k$ be the number of times arm $k$ is pulled by node $i$ up to time $t$ and let $n_t^k := \sum_{i=1}^N n_{t,i}^k$ be the number of times arm $k$ is pulled by all the nodes in the network up to time $t$. We define the regret of the whole network as

$$R(T) := TN\mu_1 - \mathbb{E}\left[\sum_{t=1}^T \sum_{i=1}^N \mu_{I_{t,i}}\right] = \sum_{k=1}^K \Delta_k \mathbb{E}\left[n_T^k\right].$$

We will use this notion of regret, which is the expected regret, in the entire paper.

The problem is to minimize the regret while allowing each agent to send poly$(K)$ values to her neighbors per iteration. We allow to know only little information about the graph. The total number of nodes and an lower bound on the spectral gap of the communication matrix $P$, i.e. $1 - |\lambda_2|$. Here $\lambda_2$ is the second greatest eigenvalue of $P$ in absolute value. The communication matrix can be build with little extra information about the graph, like the maximum degree of nodes of the graph [37]. However, building global structures is not allowed. For instance, a spanning tree to propagate the information with a message passing algorithm is not valid. This is because our focus is on designing a decentralized algorithm. Among other things, finding these kinds of decentralized solutions serves as a first step towards the design of solutions for the same problem in time varying graphs or in networks prone to communication errors.

## 3 Algorithm

We propose an algorithm that is an adaptation of UCB to the problem at hand that uses a gossip protocol. We call the algorithm Decentralized Delayed Upper Confidence Bound (DDUCB). UCB is a popular algorithm for the stochastic MAB problem. At each time step, UCB computes an upper bound of a confidence interval for the mean of each arm $k$, using two values: the empirical mean observed, $\mu_t^k$, and the number of times arm $k$ was pulled, $n_t^k$. UCB plays at time $t + 1$ the arm that maximizes the following upper confidence bound

$$\mu_t^k + \sqrt{\frac{4\eta\sigma^2 \ln t}{n_t^k}},$$

where $\eta > 1$ is an exploration parameter.

In our setting, as the pulls are distributed across the network, agents do not have access to these two values, namely the number of times each arm was pulled across the network and the empirical mean reward observed for each arm computed using the total number of pulls. Our algorithm maintains good approximations of these values and it incurs a regret that is no more than the one for a centralized

UCB plus a term depending on the spectral gap and the number of nodes, but independent of time. The latter term is a consequence of the approximation of the aforementioned values. Let $m_t^k$ be the sum of rewards coming from all the pulls done to arm $k$ by the entire network up to time $t$. We can use a gossip protocol, for every $k \in \{1, \ldots, K\}$, to obtain at each node a good approximation of $m_t^k$ and the number of times arm $k$ was pulled, i.e. $n_t^k$. Let $\widehat{m}_{t,i}^k$, $\widehat{n}_{t,i}^k$ be the approximations of $m_t^k$ and $n_t^k$ made by node $i$ with a gossip protocol at time $t$, respectively. Having this information at hand, agents could compute the ratio $\widehat{m}_{t,i}^k / \widehat{n}_{t,i}^k$ to get an estimation of the average reward of each arm. But care needs to be taken when computing the foregoing approximations.

A classical and effective way to keep a running approximation of the average of values that are iteratively added at each node is what we will refer to as the *running consensus* [8]. Let $\mathcal{N}(i)$ be the set of neighbors of agent $i$ in graph $G$. In this protocol, every agent stores her current approximation and performs communication and computing steps alternately: at each time step each agent computes a weighted average of her neighbors' values and adds to it the new value she has computed. We can represent this operation in the following way. Let $P \in \mathbb{R}^{N \times N}$ be a matrix that respects the structure of the network, which is represented by a graph $G$. So $P_{ij} = 0$ if there is no edge in $G$ that connects $j$ to $i$. We consider $P$ for which the sum of each row and the sum of each column is 1, which implies that 1 is an eigenvalue of $P$. We further assume all other eigenvalues of $P$, namely $\lambda_2, \ldots, \lambda_N$, are real and are less than one in absolute value, i.e., $1 = \lambda_1 > |\lambda_2| \geq \cdots \geq |\lambda_N| \geq 0$. Note that they are sorted by magnitude. For matrices with real eigenvalues, these three conditions hold if and only if values in the network are averaged, i.e., $P^s$ converges to $\mathbb{1}\mathbb{1}^\top / N$ for large $s$. This defines a so called gossip matrix. See [37] for a proof and [14, 37] for a discussion on how to choose $P$. If we denote by $x_t \in \mathbb{R}^N$ the vector containing the current approximations for all the agents and by $y_t \in \mathbb{R}^N$ the vector containing the new values added by each node, then the running consensus can be written as

$$x_{t+1} = P x_t + y_t. \tag{1}$$

The conditions imposed on $P$ not only ensure that values are averaged but also that the averaging process is fast. In particular, for any $s \in \mathbb{N}$ and any $v$ in the $N$-dimensional simplex

$$\|P^s v - \mathbb{1}/N\|_2 \leq |\lambda_2|^s, \tag{2}$$

see [17], for instance. For a general vector, rescale the inequality by its 1-norm. A natural approach to the problem is to use $2K$ running consensus algorithms, computing approximations of $m_t^k/N$ and $n_t^k/N$, $k = 1, \ldots, K$. Landgren et al. [25] follow this approach and use extra global information of the graph, as described in the section on related work, to account for the inaccuracy of the mean estimate. We can estimate average rewards by their ratio and the number of times each arm was pulled can be estimated by multiplying the quantity $n_t^k/N$ by $N$. The running consensus protocols would be the following. For $k = 1, \ldots, K$, start with $\widehat{m}_1^k = 0 \in \mathbb{R}^N$ and update $\widehat{m}_{t+1}^k = P\widehat{m}_t^k + \pi_t^k$, where the $i$-$th$ entry of $\pi_t^k \in \mathbb{R}^N$ contains the reward observed by node $i$ at time $t$ if arm $k$ is pulled. Else, it is 0. Note that the $i$-$th$ entry is only computed by the $i$-$th$ node. Similarly, for $k = 1, \ldots, K$, start with $\widehat{n}_1^k = 0 \in \mathbb{R}^N$ and update $\widehat{n}_{t+1}^k = P\widehat{n}_t^k + p_t^k$, where the $i$-$th$ entry of $p_t^k \in \mathbb{R}^N$ is 1 if at time $t$ node $i$ pulled arm $k$ and 0 otherwise.

The problem with this approach is that even if the values computed are being mixed at a fast pace it takes some time for the last added values to be mixed, resulting in poor approximations, especially if $N$ is large. This phenomenon is more intense when the spectral gap is smaller. Indeed, we can rewrite (1) as $x_t = \sum_{s=1}^{t-1} P^{t-1-s} y_s$, assuming that $x_1 = 0$. For the values of $s$ that are not too close to $t - 1$ we have by (2) that $P^{t-1-s} y_s$ is very close to the vector that has as entries the average of the values in $y_s$, that is, $c\mathbb{1}$, where $c = \frac{1}{N} \sum_{j=1}^N y_{s,j}$. However, for values of $s$ close to $t - 1$ this is not true and the values of $y_s$ influence heavily the resulting estimate, being specially inaccurate as an estimation of the true mean if $N$ is large. The key observations that lead to the algorithm we propose are that the number of these values of $s$ *close to* $t - 1$ is small, that we can make it even smaller using accelerated gossip techniques and that the regret of UCB does not increase much when working with delayed values of rewards so we can temporarily ignore the recently computed rewards in order to work with much more accurate approximations of $m_t^k/N$ and $n_t^k/N$. In particular, with $C$ communication steps agents can compute a polynomial $q_C$ of degree $C$ of the communication matrix $P$ applied to a vector, that is, $q_C(P)v$. The acceleration comes from computing a rescaled Chebyshev polynomial and it is encapsulated in the following lemma. It is the same one can find in previous works [30]. See the supplementary material for a proof and for the derivation of Algorithm 2 that computes $(q_C(P)v)_i$ iteratively after $C$ calls.

**Lemma 3.1.** *Let $P$ be a communication matrix with real eigenvalues such that $\mathbb{1}^\top P = \mathbb{1}^\top$, $P\mathbb{1} = \mathbb{1}$ and whose second largest eigenvalue in absolute value is $-1 < \lambda_2 < 1$. Let $v$ be in the $N$-dimensional simplex and let $C = \lceil \ln(2N/\varepsilon)/\sqrt{2\ln(1/|\lambda_2|)} \rceil$. Agents can compute, after $C$ communication steps, a polynomial $q_C$ of degree $C$ which satisfies $\|q_C(P)v - \mathbb{1}/N\|_2 \le \varepsilon/N$.*

Given the previous lemma, we consider that any value that has been computed since at least $C$ iterations before the current time step is mixed enough to be used to approximate $m_t^k/N$ and $n_t^k/N$.

We now describe DDUCB at node $i$. The pseudocode is given in Algorithm 1. We use Greek letters to denote variables that contain rewards estimators, and corresponding Latin letters to denote variables that contain counter estimators. A notation chart can be found in the supplementary material. Agents run an accelerated running consensus in stages of $C$ iterations. Each node maintains three pairs of K-dimensional vectors. The variable $\alpha_i$ contains rewards that are mixed, $\beta_i$ contains rewards that are being mixed and $\gamma_i$ contains rewards obtained in the current stage. The vectors $a_i$, $b_i$ and $c_i$ store the number of arm pulls associated to the quantities $\alpha_i$, $\beta_i$ and $\gamma_i$, respectively. At the beginning, agent $i$ pulls each arm once and initialize $\alpha_i$ and $a_i$ with the observed values divided by $N$. During each stage, for $C$ iterations, agent $i$ uses $\alpha_i$ and $a_i$, as updated at the end of the previous stage, to decide which arm to pull using an upper confidence bound. Variables $\beta_i$ and $b_i$ are mixed in an accelerated way and $\gamma_i$ and $c_i$ are added new values obtained by the new pulls done in the current stage. After $C$ iterations, values in $\beta_i$ and $b_i$ are mixed enough so we add them to $\alpha_i$ and $a_i$. The only exception being the end of the first stage in which the values of the latter variables are overwritten by the former ones. Variables $\delta_i$ and $d_i$ just serve to make this distinction. The unmixed information about the pulls obtained in the last stage, i.e. $\gamma_i$ and $c_i$, is assigned to $\beta_i$ and $b_i$ so the process can start again. Variables $\gamma_i$ and $c_i$ are reset with zeroes. There are $T$ iterations in total.

---

**Algorithm 1** DDUCB at node $i$.

1: $\zeta_i \leftarrow (X_i^1, \ldots, X_i^K)$ ; $z_i \leftarrow (1, \ldots, 1)$
2: $C = \lceil \ln(2N/\varepsilon)/\sqrt{2\ln(1/|\lambda_2|)} \rceil$
3: $\alpha_i \leftarrow \zeta_i/N$ ; $a_i \leftarrow z_i/N$ ; $\beta_i \leftarrow \zeta_i$ ; $b_i \leftarrow z_i$
4: $\gamma_i \leftarrow \mathbf{0}$ ; $c_i \leftarrow \mathbf{0}$ ; $\delta_i \leftarrow \mathbf{0}$ ; $d_i \leftarrow \mathbf{0}$
5: $t \leftarrow K$ ; $s \leftarrow K$
6: **while** $t \le T$ **do**
7:   $k^* \leftarrow \arg\max_k \left\{ \frac{\alpha_i^k}{a_i^k} + \sqrt{\frac{2\eta\sigma^2 \ln s}{N a_i^k}} \right\}$
8:   **for** $r$ from 0 to $C-1$ **do**
9:     $u \leftarrow$ Play arm $k^*$, return reward
10:    $\gamma_i^{k^*} \leftarrow \gamma_i^{k^*} + u$ ; $c_i^{k^*} \leftarrow c_i^{k^*} + 1$
11:    $\beta_i \leftarrow \text{mix}(\beta_i, r, i)$ ; $b_i \leftarrow \text{mix}(b_i, r, i)$
12:    $t \leftarrow t + 1$
13:    **if** $t > T$ **then** return **end if**
14:   **end for**
15:   $s \leftarrow (t - C)N$
16:   $\delta_i \leftarrow \delta_i + \beta_i$ ; $d_i \leftarrow d_i + b_i$ ; $\alpha_i \leftarrow \delta_i$ ; $a_i \leftarrow d_i$
17:   $\beta_i \leftarrow \gamma_i$ ; $b_i \leftarrow c_i$ ; $\gamma_i \leftarrow \mathbf{0}$ ; $c_i \leftarrow \mathbf{0}$
18: **end while**

---

**Algorithm 2** Accelerated communication and mixing step. $\text{mix}(y_{r,i}, r, i)$

1: **if** $r$ is 0 **then**
2:   $w_0 \leftarrow 1/2$ ; $w_{-1} \leftarrow 0$
3:   $y_{0,i} \leftarrow y_{0,i}/2$ ; $y_{-1,i} \leftarrow (0, \ldots, 0)$
4: **end if**
5: Send $y_{r,i}$ to neighbors
6: Receive corresp. values $y_{r,j}, \forall j \in \mathcal{N}(i)$
7: $y'_{r,i} \leftarrow \sum_{j \in \mathcal{N}(i)} 2P_{ij} y_{r,j} / |\lambda_2|$
8: $w_{r+1} \leftarrow 2w_r/|\lambda_2| - w_{r-1}$
9: $y_{r+1,i} = \frac{w_r}{w_{r+1}} y'_{r,i} - \frac{w_{r-1}}{w_{r+1}} y_{r-1,i}$
10: **if** $r$ is 0 **then**
11:   $y_{0,i} \leftarrow 2y_{0,i}$ ; $w_0 \leftarrow 2w_0$
12: **end if**
13: return $y_{r+1,i}$

---

Now we describe some mathematical properties about the variables during the execution of the algorithm. Let $t_S$ be the time at which a stage begins, so it ends at $t_S + C - 1$. At $t = t_S$, using the notation above, it is $\alpha_i^k = \sum_{s=1}^{t_S - C} \left( q_C(P)\pi_s^k \right)_i$ and $a_i^k = \sum_{s=1}^{t_S - C} \left( q_C(P)p_s^k \right)_i$ but in the first stage, in which their values are initialized from a local pull. In particular, denote $X_i^1, \ldots, X_i^K$ the rewards obtained when pulling all the arms before starting the first stage. Then the initialization is $\alpha_i \leftarrow (X_i^1/N, \ldots, X_i^K/N)$ and $a_i \leftarrow (1/N, \ldots, 1/N)$. The division by $N$ is due to $\alpha_i^k$ and $a_i^k$ being the approximations for $m_{t,i}^k/N$ and $n_{t,i}^k/N$. The algorithm does not update $\alpha_i$ and $a_i$ again until $t = t_S + C$, so they contain information that at the end of the stage is delayed by $2C - 1$ iterations. The time step $s$ used to compute the upper confidence bound is $(t_S - C)N$, since $\alpha_i$ and $a_i$ contain information about that number of rewards and pulls. The variable $\gamma_i$ is needed because we need to mix $\beta_i$ for $C$ steps so the Chebyshev polynomial of degree $C$ is computed. In this way agents compute upper confidence bounds with accurate approximations, with a delay of at most $2C - 1$.

As we will see, the regret of UCB does not increase much when working with delayed estimates. In particular, having a delay of $\mathfrak{d}$ steps increases the regret by at most $\mathfrak{d} \sum_{k=1}^{K} \Delta_k$.

We now present the regret which the DDUCB algorithm incurs. We use $A \lesssim B$ to denote there is a constant $c > 0$ such that $A \le cB$. See Appendix A.1 for a proof.

**Theorem 3.2 (Regret of DDUCB).** *Let $P$ be a communication matrix with real eigenvalues such that $\mathbb{1}^\top P = \mathbb{1}^\top$, $P\mathbb{1} = \mathbb{1}$ whose second largest eigenvalue in absolute value is $\lambda_2$, with $|\lambda_2| < 1$. Consider the distributed multi-armed bandit problem with $N$ nodes, $K$ actions and subgaussian rewards with variance proxy $\sigma^2$. The algorithm DDUCB with exploration parameter $\eta = 2$ and $\varepsilon = 1/22$ satisfies:*

1. *The following finite-time bound on the regret, for $C = \lceil \frac{\log(2N/\varepsilon)}{\sqrt{2\ln(1/|\lambda_2|)}} \rceil$*

$$ R(T) < \sum_{k:\Delta_k > 0} \frac{32(1 + 1/11)\sigma^2 \ln(TN)}{\Delta_k} + \left( N(6C + 1) + 4 \right) \sum_{k=1}^{K} \Delta_k. $$

2. *The corresponding asymptotic bound:*

$$ R(T) \lesssim \sum_{k:\Delta_k > 0} \frac{\sigma^2 \ln(TN)}{\Delta_k} + \frac{N\ln(N)}{\sqrt{\ln(1/|\lambda_2|)}} \sum_{k=1}^{K} \Delta_k. $$

For simplicity and comparison purposes we set the value of $\eta$ and $\varepsilon$ to specific values. For a general version of Theorem 3.2, see the supplementary material. Note that the algorithm needs to know $\lambda_2$, the second largest eigenvalue of $P$ in absolute value, since it is used to compute $C$, which is a parameter that indicates when values are close enough to be mixed. However, if we use DDUCB with $C$ set to any upper bound $E$ of $C = \lceil \log(2N/\varepsilon)/\sqrt{2\ln(1/|\lambda_2|)} \rceil$ the inequality of the finite-time analysis above still holds true, substituting $C$ by $E$. In the asymptotic bound, $N\ln N/\sqrt{\ln(1/|\lambda_2|)}$ would be substituted by $NE$. The knowledge of the spectral gap is an assumption that is widely made throughout the decentralized literature [13, 14, 30]. We can use Theorem 3.2 to derive an instance-independent analysis of the regret. See Theorem A.3 in the supplementary material.

**Remark 3.3 (Lower bound).** In order to interpret the regret obtained in the previous theorem, it is useful to note that running the centralized UCB algorithm for $TN$ steps incurs a regret bounded above by $\sum_{k:\Delta_k > 0} \frac{\sigma^2 \ln(TN)}{\Delta_k} + \sum_{k=1}^{K} \Delta_k$, up to a constant. Moreover, running $N$ separate instances of UCB at each node without allowing communication incurs a regret of $R(T) \lesssim \sum_{k:\Delta_k > 0} \frac{N\sigma^2 \ln(T)}{\Delta_k} + N \sum_{k=1}^{K} \Delta_k$. On the other hand, the following is an asymptotic lower bound for any consistent centralized policy [23]: $\liminf_{T \to \infty} \frac{R(T)}{\ln T} \ge \sum_{k:\Delta_k > 0} \frac{2\sigma^2}{\Delta_k}$.

Thus, we see that the regret obtained in Theorem 3.2 improves significantly the dependence on $N$ of the regret with respect to the trivial algorithm that does not involve communication, and that it is asymptotically optimal in terms of $T$, with $N$ and $K$ fixed. Since in the first iteration of this problem $N$ arms have to be pulled and there is no prior information on the arms' distribution, any asymptotically optimal algorithm in terms of $N$ and $K$ must pull $\Theta(\frac{N}{K} + 1)$ times each arm, yielding regret of at least $\left( \frac{N}{K} + 1 \right) \sum_{k=1}^{K} \Delta_k$, up to a constant. Hence, by the lower bound above and the latter argument, we can give the following lower bound for the problem we consider. The regret of our problem must be

$$ \Omega\left( \sum_{k:\Delta_k > 0} \frac{\sigma^2 \ln(TN)}{\Delta_k} + \left( \frac{N}{K} + 1 \right) \sum_{k=1}^{K} \Delta_k \right), $$

and the regret obtained in Theorem 3.2 is asymptotically optimal up to at most a factor of $\min(K, N)\ln(N)/\sqrt{\ln(1/|\lambda_2|)}$ in the second summand of the regret.

**Remark 3.4 (Comparison with previous work).** We note that in [25] the regret bounds were computed applying a concentration inequality that cannot be used, since it does not take into account that the number of times an arm was pulled is a random variable. They claim their analysis follows the one in [4], which does not present this problem. If we changed their upper confidence bound

to be proportional to $\sqrt{6\ln(tN)}$ instead of to $\sqrt{2\ln(t)}$ at time $t$ and follow [4] then, for their best algorithm in terms of regret, named coopUCB, we can get a very similar regret bound to the one they obtained. The regret of their algorithm is bounded by $A + B\sum_{k=1}^{K}\Delta_k$, where

$$A := \sum_{k:\Delta_k>0}\sum_{j=1}^{N}\frac{16\gamma\sigma^2(1+\varepsilon_c^j)}{N\Delta_k}\ln(TN), \quad B := N\Big(\frac{\gamma}{\gamma-1}+\sqrt{N}\sum_{j=2}^{N}\frac{|\lambda_j|}{1-|\lambda_j|}\Big).$$

The difference between this bound and the one presented in [25] is the $N$ inside the logarithm in $A$ and a factor of 2 in $A$. Here, $\gamma > 1$ is an exploration parameter that the algorithm receives as input and $\varepsilon_c^j$ is a non-negative graph-dependent value, which is only 0 when the graph is the complete graph. Thus $A$ is at least $\sum_{k:\Delta_k>0}\frac{16\sigma^2\ln(TN)}{\Delta_k}$. Hence, up to a constant, $A$ is always greater than the first summand in the regret of our algorithm in Theorem 3.2. Note that $\frac{\gamma}{\gamma-1}\geq 1$ and $\frac{1}{1-|\lambda_2|}\geq\frac{1}{\ln(|\lambda_2|^{-1})}$ so

$$B \geq N\left(1+\frac{\lambda_2'}{\ln(\sqrt{N}/\lambda_2')}\right),$$

where $\lambda_2' := \sqrt{N}|\lambda_2| \in [0,\sqrt{N})$. The factor multiplying $\sum_{k=1}^{K}\Delta_k$ in the second summand in Theorem 3.2 is $N\ln N/\sqrt{\ln(1/|\lambda_2|)} \leq N\ln N/\ln(1/|\lambda_2|) \leq 2B$, for $|\lambda_2| \geq 1/e$, since the inequality below holds.

$$2B \geq 2N\Big(1+\frac{\lambda_2'}{\ln(\sqrt{N}/\lambda_2')}\Big) \geq \frac{N\ln N}{\ln(\sqrt{N}/\lambda_2')} \Leftrightarrow \ln N - 2\ln(\lambda_2') + 2\lambda_2' \geq \ln N.$$

See the case $|\lambda_2| < 1/e$ in Appendix D. In the case of a complete graph, the problem reduces to a centralized batched bandit problem, in which N actions are taken at each time step [29]. The communication in this case is trivial, just send the obtained rewards to your neighbors, so not surprisingly our work and [25] incur the same regret in such a case. The previous reasoning proves, however, that for every graph our asymptotic regret is never worse and for many graphs we get substantial improvement. Depending on the graph, $A$ and $B$ can be much greater than the lower bound we have used for both of them for comparison purposes. In the supplementary material, for instance, we show that in the case of a cycle graph with a natural communication matrix these two parts are substantially worse in [25], namely $\Theta(N^2)$ versus $\Theta(1)$ and $\Theta(N^{7/2})$ versus $\Theta(N^2\log N)$ for the term multiplying $\sum_{k:\Delta_k>0}\sigma^2\ln(TN)/\Delta_k$ in $A$ and for $B$, respectively. In general, the algorithm we propose presents several improvements. We get a graph-independent value multiplying $\ln(TN)$ in the first summand of the regret whereas $A$ contains the $1+\varepsilon_c^j$ graph-dependent values. In $B$, just the sum $N(\frac{\gamma}{\gamma-1}+\sqrt{N}\frac{|\lambda_2|}{1-|\lambda_2|})$ is of greater order than our second summand. Moreover, $B$ contains other terms depending on the eigenvalues $\lambda_j$ for $j \geq 3$. Furthermore, we get this while using less global information about the graph. This is of interest for decentralization purposes. Note however it has computational implications as well, since in principle the computation of $\varepsilon_c^j$ needs the entire set of eigenvalues and eigenvectors of $P$. Thus, even if $P$ were input to coopUCB, it would need to run an expensive procedure to compute these values before starting executing the decision process, while our algorithm does not need.

**Remark 3.5 (Variants of DDUCB).** The algorithm can be modified slightly to obtain better estimations of $m_t^k/N$ and $n_t^k/N$, which implies the regret is improved. The easiest (and recommended) modification is the following. While waiting for the vectors $\beta_i$ and $b_i$, $i = 1,\dots N$ to be mixed, each node $i$ adds to the variables $\alpha_i$ and $a_i$ the information of the pulls that are done times $1/N$. The variable $s$ accounting for the time step has to be modified accordingly. It contains the number of pulls made to obtain the approximations of $\alpha_i$ and $a_i$, so it needs to be increased by one when adding one extra reward. This corresponds to uncommenting lines 14-15 in Algorithm 5, the pseudocode in the supplementary material. Since the values of $\alpha_i$ and $a_i$ are overwritten after the for loop, the assignment of $s$ after the loop remains unchanged. Note that if the lines are not uncommented then each time the for loop is executed the $C$ pulls that are made in a node are taken with respect to the same arm. Another variant that would provide better estimations and therefore better regret, while keeping the communication cost $O(K)$ would consist of also sending the information of the new pull, $\pi_i$ and $p_i$, to the neighbors of $i$, receiving their respective values of their new pulls and adding these values to $\alpha_i$ and $a_i$ multiplied by $1/N$, respectively. We analyze the algorithm without any modification for the sake of clarity of exposition. The same asymptotic upper bound on the regret in Theorem 3.2 can be computed for these two variations.

We can vary the communication rate with some trade-offs. On the one hand, we can mix values of $\delta_i$ and $d_i$ at each iteration of the for loop, in an unaccelerated way and with Algorithm 3 (see Algorithm 5, line 17 in the supplementary material), to get even more precise estimations. In such a case, we could use $\delta_i$ and $d_i$ to compute the upper confidence bounds instead of $\alpha_i$ and $a_i$. However, that approach cannot benefit from using the information from local pulls obtained during the stage. On the other hand, if each agent could not communicate $2K$ values per iteration, corresponding to the mixing in line 11, the algorithm can be slightly modified to account for it at the expense of incurring greater regret. Suppose each agent can only communicate $L$ values to her neighbors per iteration. Let $E$ be $\lceil 2KC/L \rceil$. If each agent runs the algorithm in stages of $E$ iterations, ensuring to send each element of $\beta_i$ and $b_i$ exactly $C$ times and using the mixing step $C$ times, then the bounds in Theorem 3.2, substituting $C$ by $E$, still hold. Again, in the asymptotic bound, $N \ln N / \sqrt{\ln(1/|\lambda_2|)}$ would be substituted by $NE$. In each iteration, agents have to send values corresponding to the same entries of $\beta_i$ or $b_i$. The factor of $C$ in the second summand of the regret accounts for the number of rounds of delay since a reward is obtained until it is used to compute upper confidence bounds. If we decrease the communication rate and compensate it with a greater delay, the approximations in $\alpha_i$ and $a_i$ satisfy the same properties as in the original algorithm. Only the second summand in the regret increases because of an increment of the delay.

**Experiments.** We show that the algorithm proposed in this work, DDUCB, does not only enjoy a better theoretical regret guarantee but it also performs better in practice. In general we have observed that the accelerated method performs well with the recommended values, that is, no tuning, for the exploration parameter $\eta$ and the parameter $\varepsilon$ that measures the precision of the mixing after a stage. Remember these values are $\eta = 2$, $\varepsilon = \frac{1}{22}$. On the other hand the constant $C$ that results in the unaccelerated method is usually excessively large, so it is convenient to heuristically decrease it, which corresponds to using a different value of $\varepsilon$. We set $\varepsilon$ so the value of $C$ for the unaccelerated method is the same as the value of $C$ for the accelerated one. We have used the recommended modification of DDUCB consisting of adding to the variables $\alpha_i$ and $a_i$ the information of the pulls that are done times $1/N$ while waiting for the vectors $\beta_i$ and $b_i$ to be mixed. This modification adds extra information that is at hand at virtually no computational cost so it is always convenient to use it.

We tuned $\gamma$, the exploration parameter of coopUCB [25], to get best results for that algorithm and plot the executions for the best $\gamma$'s and also $\gamma = 2$ for comparison purposes. In the figures one can observe that after a few stages, DDUCB algorithms learn with high precision which the best arm is and the regret curve that is observed afterwards shows an almost horizontal behavior. After 10000 iterations, coopUCB not only accumulates a greater regret but the slope indicates that it still has not learned effectively which the best arm is.

See Appendix G for a more detailed description about the experiments.

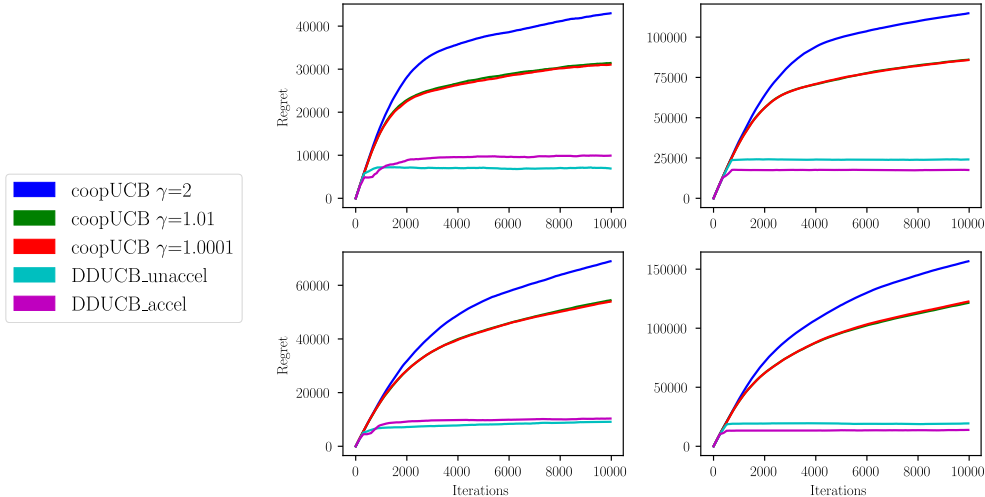

Figure 1: Simulation of DDUCB and coopUCB for cycles (top) and square grids (bottom) for 100 nodes (left) , 200 nodes (top right) and 225 nodes (bottom right).

**Acknowledgments**

The authors thank Raphaël Berthier and Francis Bach for helpful exchanges on the problem of averaging. David Martínez-Rubio was supported in part by EP/N509711/1 from the EPSRC MPLS division, grant No 2053152. Varun Kanade and Patrick Rebeschini were supported in part by the Alan Turing Institute under the EPSRC grant EP/N510129/1. The authors acknolwedge support from the AWS Cloud Credits for Research program.

## Footnotes

[1]A high-level description of some distributed algorithms is given in the related work section. For further details, the reader is referred to the references in that section.

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
