[Supplementary Material]

# A  Proofs

## A.1  Proof of Theorem 3.2

The proof is along the lines of the one for the standard UCB algorithm cf. [4] but requires a couple of key modifications. Firstly, we need to control the error due to the fact that each agent decides which arm to pull with some delay, because only information after it is mixed is used. Secondly, we need to control the error due to the fact that agents only have approximations of $m_t^k$ and $n_t^k$, that is, to the true sum of rewards and number of times each arm was pulled respectively.

We present two lemmas before the proof. Their proofs can be found in Appendix A.2. Note the running consensus operation is linear. The linearity of $P$ allows us to think about each reward as being at each node weighted by a number. For each reward, since $P$ is a gossip matrix the sum of the weights across all the nodes is $1$ and the weights approach $1/N$ quickly.

**Lemma A.1.** *Fix an arm $k$, a node $i$, and a time step $t$. Let $Y_1, \ldots, Y_D$ be independent random variables coming from the distribution associated to arm $k$, which we assume subgaussian with variance proxy $\sigma^2$ and mean $\mu_k$. Let $\varepsilon > 0$ be arbitrarily small and less than $\frac{\eta-1}{7(\eta+1)}$, for $\eta > 1$, and let $s > 1$. Let $w_j$ be a number such that $|w_j - \frac{1}{N}| < \varepsilon/N$, where $j = 1, \ldots, D$. Then*

$$\mathbb{P}\left[\frac{\sum w_j Y_j}{\sum w_j} \geq \mu_k + \sqrt{\frac{4\eta\sigma^2 \ln s}{N \sum w_j}}\right] \leq \frac{1}{s^{\eta+1}} \;\; and \;\; \mathbb{P}\left[\frac{\sum w_j Y_j}{\sum w_j} \leq \mu_k - \sqrt{\frac{4\eta\sigma^2 \ln s}{N \sum w_j}}\right] \leq \frac{1}{s^{\eta+1}},$$

*where the sums go from $j = 1$ to $j = D$.*

At time $t$ and at node $i$, we want to use the variables $\alpha_i$ and $a_i$ defined in Algorithm 1 to decide the next arm to pull in that node. Consider the rewards computed by all the nodes until $C$ steps before the last time $\alpha_i$ and $a_i$ were updated. Let $J_t^k$ be the number of these rewards that come from arm $k$ and let $X_j^k, 1 \leq j \leq J_t^k$ be such rewards. We can see each of the $X_j^k$ as being at node $i$ at the beginning of iteration $t$ multiplied by a weight $w_{t,i,j}^k$. Every weight we are considering corresponds to a reward that has been mixing for at least $C = \lceil \frac{\ln(2N/\varepsilon)}{\sqrt{2\ln|\lambda_2|^{-1}}} \rceil$ steps. This ensures $\left|w_{t,i,j}^k - \frac{1}{N}\right| < \frac{\varepsilon}{N}$ by Lemma 3.1, so the previous lemma can be applied to these weights.

Define the empirical mean of arm $k$ at node $i$ and time $t$ as

$$\widehat{\mu}_{t,i}^k := \frac{\sum_{j=1}^{J_t^k} w_{t,j,i}^k X_j^k}{\sum_{j=1}^{J_t^k} w_{t,j,i}^k},$$

Let $\text{UCB}(t,s,k,i) := \widehat{\mu}_{t,i}^k + \sqrt{\frac{4\eta\sigma^2 \ln s}{N \sum_j w_{t,i,j}^k}}$ and let $I_{t,i}$ be the random variable that represents the arm pulled at time $t$ by node $i$, which is the one that maximizes $\text{UCB}(t,s,k,i)$, for a certain value $s$.

**Lemma A.2.** *Let $k^*$ and $k$ be an optimal arm and a suboptimal arm respectively. We have*

$$\mathbb{P}\left(I_{t,i} = k, N \sum_j w_{t,i,j}^k > \frac{16\eta\sigma^2 \ln s}{\Delta_k^2}\right) \leq \frac{2Ns_t}{s^{\eta+1}}.$$

*where $Ns_t = \sum_{k=1}^{K} J_k^t$ is the number of rewards obtained by all the nodes until $C$ steps before the last time $\alpha_i$ and $a_i$ were updated.*

Now we proceed to prove the theorem.

*Proof* (Theorem 3.2). For every $t \geq K$ we can write $t$ uniquely as $K + Cq_t + r_t$, where $q_t \geq 0$ and $0 \leq r_t < C$. In such a case it is

$$s_t = K \max\left(\mathbb{1}(q_t > 0), 1/N\right) + C(q_t - 1)\mathbb{1}(q_t > 1),$$

where $s_t$ is defined in Lemma A.2. The time step $s$ that we use to compute the upper confidence bounds at time $t$ is $s = Ns_t$. It is fixed every $C$ iterations. For $t \geq K + C$, the value $s_t + C$ is equal to the last time step in which the variables $\alpha_i$ and $a_i$ were updated. Thus by definition $J_t^k = n_{s_t}^k$. Remember $n_{t,i}^k$ is the number of times arm $k$ is pulled by node $i$ up to time $t$, and $n_t^k = \sum_{i=1}^{N} n_{t,i}^k$. Since $R(T) = \sum_{k=1}^{K} \Delta_k \mathbb{E}[n_T^k]$ it is enough to bound $\mathbb{E}[n_T^k]$ for every $k = 1, \ldots, K$.

Let $k$ be fixed and denote $A_{t,i}$ the event $\{I_{t,i} = k\}$. We have

$$\mathbb{E}[n_T^k] = N + \mathbb{E}\left[\sum_{i=1}^{N}\sum_{t=K+1}^{T} \mathbb{1}(A_{t,i})\right]$$

$$= N + \mathbb{E}\left[\sum_{i=1}^{N}\sum_{t=K+1}^{T} \mathbb{1}\left(A_{t,i}, 1 \leq \frac{16\eta\sigma^2 \ln(s_t N)}{N\sum_j w_{t,i,j}^k \Delta_k^2}\right) + \mathbb{1}\left(A_{t,i}, 1 > \frac{16\eta\sigma^2 \ln(s_t N)}{N\sum_j w_{t,i,j}^k \Delta_k^2}\right)\right]$$

$$\overset{\textcircled{1}}{\leq} N + \mathbb{E}\left[\sum_{i=1}^{N}\sum_{t=K+1}^{T} \mathbb{1}\left(A_{t,i}, n_{s_t}^k \leq \frac{16\eta\sigma^2 \ln(TN)}{\Delta_k^2/(1+2\varepsilon)}\right)\right] + N\sum_{t=K+1}^{T}\frac{2}{(s_t N)^\eta}$$

$$\overset{\textcircled{2}}{<} N + \frac{16\eta\sigma^2 \ln(TN)}{\Delta_k^2/(1+2\varepsilon)} + 2NC + \frac{2NC}{K^\eta}\left(1 + \frac{1}{N^\eta}\right) + \frac{2}{(NC)^{\eta-1}}\sum_{r=\lfloor K/C\rfloor+1}^{\infty}\frac{1}{r^\eta}$$

$$\overset{\textcircled{3}}{\lesssim} \frac{\eta\sigma^2 \ln(TN)}{\Delta_k^2/(1+\varepsilon)} + \frac{N\ln(N/\varepsilon)}{\sqrt{\ln(1/|\lambda_2|)}} + \frac{\eta}{\eta-1}.$$

For the bound of the first summand in $\textcircled{1}$ note that $\left|w_{t,i,j}^k - \frac{1}{N}\right| < \varepsilon/N$ and that $J_t^k = n_{s_t}^k$. Thus

$$\left(N\sum_j w_{t,i,j}^k\right)^{-1} \leq \left(n_{s_t}^k (1-\varepsilon)\right)^{-1} \leq (1+2\varepsilon)/n_{s_t}^k.$$

We have used $\varepsilon < 1/2$ for the last step, which is a consequence of $\varepsilon < \frac{\eta-1}{7(\eta+1)}$ for $\eta > 1$. The bound for the second summand uses Lemma A.2. For the bound of the expectation in $\textcircled{2}$, note that, by definition, $A_{t,i}$ for $1 \leq t \leq T$ can only happen $n_t^k$ times but

$$n_t^k \leq n_{s_t}^k + N(t - s_t) \leq n_{s_t}^k + 2NC.$$

So $\mathbb{1}\left(I_{t,i+1} = k, n_{s_t}^k < \frac{8\eta\sigma^2 \ln(TN)}{\Delta_k^2/(1+2\varepsilon)}\right)$ can be 1 at most $\frac{8\eta\sigma^2 \ln(TN)}{\Delta_k^2/(1+2\varepsilon)} + 2NC$ times. The term $2NC$ accounts for the delay of the algorithm. In the second part of inequality $\textcircled{2}$ we substitute $s_t$ by its value and for $t > K + 2C$ we bound it by the greatest multiple of $C$ that is less than $s_t$. For $\textcircled{3}$, note that the sum over $r$ is bounded by $\zeta(\eta)$, where $\zeta(\cdot)$ is the Riemann zeta function. Then we use $\zeta(x) < \frac{x}{x-1}$ for all $x > 1$, cf. [18], Proposition 16.1.2. Substituting the value of $C$ and the values $\eta = 2, \varepsilon = 1/22$ then yields the bound. The finite-time bound follows by bounding the Riemann zeta function as above and bounding $1/N^\eta$, $1/K^\eta$ and $1/(NC)^{\eta-1}$ by 1. $\qquad\square$

## A.2 Other proofs

**Lemma 3.1.** *Let $P$ be a communication matrix with real eigenvalues such that $\mathbb{1}^\top P = \mathbb{1}^\top$, $P\mathbb{1} = \mathbb{1}$ and whose second largest eigenvalue in absolute value is $-1 < \lambda_2 < 1$. Let $v$ be in the $N$-dimensional simplex and let $C = \lceil \ln(2N/\varepsilon)/\sqrt{2\ln(1/|\lambda_2|)}\rceil$. Agents can compute, after $C$ communication steps, a polynomial $q_C$ of degree $C$ which satisfies*

$$\|q_C(P)v - \mathbb{1}/N\|_2 \leq \varepsilon/N.$$

*Proof.* Define the Chebyshev polynomials as $T_0(t) = 1$, $T_1(t) = t$ and $T_r(t) = 2tT_{r-1}(t) - T_{r-2}(t)$ for $r > 1$. Then, define

$$q_r(t) = \frac{T_r(t/|\lambda_2|)}{T_r(1/|\lambda_2|)}.$$

Let $\kappa = \frac{1+|\lambda_2|}{1-|\lambda_2|}$, and $C = \left\lceil \frac{\ln(2N/\varepsilon)}{\sqrt{2\ln(1/|\lambda_2|)}}\right\rceil$. Then for any $t \in [-|\lambda_2|, |\lambda_2|]$ the polynomial $q_C$ satisfies:

$$q_C(t) \overset{\textcircled{1}}{\leq} 2\frac{\left(\frac{\sqrt{\kappa}-1}{\sqrt{\kappa}+1}\right)^C}{1 + \left(\frac{\sqrt{\kappa}-1}{\sqrt{\kappa}+1}\right)^{2C}} < 2\left(\frac{\sqrt{\kappa}-1}{\sqrt{\kappa}+1}\right)^C \overset{\textcircled{2}}{\leq} 2\exp\left(-\log(2N/\varepsilon)\right) \leq \frac{\varepsilon}{N}.$$

See [5] for ①. Inequality ② is true since $\left(1 + \frac{-2}{\sqrt{(1+x)/(1-x)}+1}\right)^{1/\sqrt{2\ln(1/x)}} \leq e^{-1}$ for $x \in [0,1)$, because the expression is monotone and the $\lim_{x\to 1^-}$ of it is $e^{-1}$. It is also $q_C(1) = 1$. This implies that the absolute value of all the eigenvalues of the matrix $q_C(P)$ is less than $\frac{\varepsilon}{N}$ but for the greatest, which is 1. The previous property implies $\left\|q_C(P) - \frac{1}{N}\mathbb{1}\mathbb{1}^\top\right\|_2 \leq \frac{\varepsilon}{N}$, see [37]. Alternatively, the latter can be proven easily if $q_C(P) - \frac{1}{N}\mathbb{1}\mathbb{1}^\top$ is diagonalizable and then the result can be straightforwardly extended to all matrices since the property is continuous and the set of diagonalizable matrices is dense. Finally, for $v$ in the $N$-dimensional simplex we have

$$\left\|q_C(P)v - \mathbb{1}/N\right\|_2 = \left\|q_C(P)v - \frac{1}{N}\mathbb{1}\mathbb{1}^\top v\right\|_2 \leq \left\|q_C(P) - \frac{1}{N}\mathbb{1}\mathbb{1}^\top\right\|_2 \|v\|_2 \leq \frac{\varepsilon}{N}$$

Note that the polynomial $q_r(P)$ can be computed iteratively as

$$w_{r+1}q_{r+1}(P) = \frac{2}{|\lambda_2|}w_r P q_r(P) - w_{r-1}q_{r-1}(P), \tag{3}$$

for $r \geq 1$ where $w_r = T_r(1/|\lambda_2|)$. By the properties of the Chebyshev polynomial, $w_r$ can be computed iteratively as $w_0 = 1$, $w_1 = 1/|\lambda_2|$ and $w_{r+1} = 2w_r/|\lambda_2| - w_{r-1}$ for $r > 1$.

Also note that if we have a vector $u \in \mathbb{R}^N$ we can slightly modify the recursion in Equation (3) to compute $q_C(P)u$ using the gossip protocol $C$ times:

$$y_{r+1} = \frac{w_r}{w_{r+1}}\frac{2}{|\lambda_2|}Py_r - \frac{w_{r-1}}{w_{r+1}}y_{r-1}$$

for $r > 1$, where we denote $y_r = q_r(P)u \in \mathbb{R}^N$. In order to simplify the code we want to allow the computation of $y_1$ with the same recursion. In such a case we can use by convention $w_{-1} = 0$, $y_{-1} = 0$ but also we need to set $w_0 = 1/2$ and $y_0 = u/2$ temporarily during the computation of $y_1$, the first iteration, instead of having $w_0 = 1$ and $y_0 = u$. Pseudocode for an iteration of this recursion at node $i$ is provided in Algorithm 2, in which we have called $y_r' = 2/|\lambda_2|Py_r$. The multiplication by $P$ is performed using the gossip protocol. Agent $i$ only computes the $i$-th entry of $y_{r+1}$, namely $y_{r+1,i}$, and only uses her local information and the entries of $y_r$ corresponding to her neighbors. □

**Lemma A.1.** *Fix an arm $k$, a node $i$, and a time step $t$. Let $Y_1, \ldots, Y_D$ be independent random variables coming from the distribution associated to arm $k$, which we assume subgaussian with variance proxy $\sigma^2$ and mean $\mu_k$. Let $\varepsilon > 0$ be arbitrarily small and less than $\frac{\eta-1}{7(\eta+1)}$, for $\eta > 1$, and let $s > 1$. Let $w_j$ be a number such that $|w_j - \frac{1}{N}| < \varepsilon/N$, where $j = 1, \ldots, D$. Then*

$$\mathbb{P}\left[\frac{\sum w_j Y_j}{\sum w_j} \geq \mu_k + \sqrt{\frac{4\eta\sigma^2 \ln s}{N \sum w_j}}\right] \leq \frac{1}{s^{\eta+1}} \quad \text{and} \quad \mathbb{P}\left[\frac{\sum w_j Y_j}{\sum w_j} \leq \mu_k - \sqrt{\frac{4\eta\sigma^2 \ln s}{N \sum w_j}}\right] \leq \frac{1}{s^{\eta+1}},$$

*where the sums go from $j = 1$ to $j = D$.*

*Proof.* Since $Y_j$ is subgaussian with variance proxy $\sigma^2$ we have that $w_j Y_j/(\sum w_j)$ is subgaussian with variance proxy $w_j^2\sigma^2/(\sum w_j)^2$. Therefore, using subgaussianity and the fact that the random variables $Y_j$, for $j = 1, \ldots, D$, are independent we obtain by Hoeffding's inequality for subgaussian random variables

$$\mathbb{P}\left[\frac{\sum w_j Y_j}{\sum w_j} \geq \mu_k + \sqrt{\frac{4\eta\sigma^2 \ln s}{N \sum w_j}}\right] \leq \exp\left(-\frac{(4\eta\sigma^2 \ln s)/(N \sum w_j)}{2\sigma^2 \sum w_j^2/(\sum w_j)^2}\right) = \frac{1}{s^{2\eta/(NW)}}$$

where $W := \sum w_j^2/\sum w_j$. Using $\left|w_j - \frac{1}{N}\right| < \varepsilon/N$ we obtain

$$\eta/(NW) = \eta\left(N\frac{\sum w_j^2}{\sum w_j}\right)^{-1} \geq \eta\left(N\frac{D((1+\varepsilon)/N)^2}{D((1-\varepsilon)/N)}\right)^{-1} = \frac{\eta(1-\varepsilon)}{(1+\varepsilon)^2} > \frac{\eta+1}{2}.$$

The last step is a consequence of $\varepsilon < \frac{\eta-1}{7(\eta+1)}$. The first result follows. The second inequality is analogous. □

**Lemma A.2.** *Let $k^*$ and $k$ be an optimal arm and a suboptimal arm respectively. We have*

$$\mathbb{P}\left(I_{t,i} = k, N \sum_j w_{t,i,j}^k > \frac{16\eta\sigma^2 \ln s}{\Delta_k^2}\right) \leq \frac{2s_t N}{s^{\eta+1}}.$$

*where $N s_t = \sum_{k=1}^{K} J_k^t$ is the number of rewards obtained by all the nodes until $C$ steps before the last time $\alpha_i$ and $a_i$ were updated.*

*Proof.* It is enough to bound $\mathbb{P}\left(UCB(t, s, k^*, i) \leq \mu_{k^*}\right)$ and $\mathbb{P}\left(\widehat{\mu}_{t,i}^k > \mu_k + \sqrt{\frac{4\eta\sigma^2 \ln s}{N \sum w_j}}\right)$, since if these two events are false we can apply ① and ③ in the following and obtain $I_{t,i} \neq k$:

$$UCB(t, s, k, i) = \widehat{\mu}_{t,i}^k + \sqrt{\frac{4\eta\sigma^2 \ln s}{N \sum_j w_{t,i,j}^k}}$$

$$\stackrel{①}{\leq} \mu_k + 2\sqrt{\frac{4\eta\sigma^2 \ln s}{N \sum_j w_{t,i,j}^k}}$$

$$\stackrel{②}{<} \mu_k + \Delta_k$$

$$= \mu_{k^*}$$

$$\stackrel{③}{<} UCB(t, s, k^*, i).$$

Inequality ② is true since $N \sum_j w_{t,i,j}^k > \frac{16\eta\sigma^2 \ln s}{\Delta_k^2} \iff \sqrt{\frac{4\eta\sigma^2 \ln s}{N \sum_j w_{t,i,j}^k}} < \frac{\Delta}{2}$.

Now since $1 \leq J_k^t \leq N s_t$ we have by the union bound and Lemma A.1

$$\mathbb{P}\left(UCB(t, s, k^*, i) \leq \mu_{k^*}\right) \leq \mathbb{P}\left(\exists \ell \in \{1, \ldots, N s_t\} : J_k^t = \ell, UCB(t, s, k^*, i) \leq \mu_{k^*}\right)$$

$$\leq \sum_{\ell=1}^{N s_t} \mathbb{P}\left(UCB(t, s, k^*, i) \leq \mu_{k^*} | J_k^t = \ell\right)$$

$$\leq \sum_{\ell=1}^{N s_t} \frac{1}{s^{\eta+1}} = \frac{N s_t}{s^{\eta+1}}.$$

The bound of $\mathbb{P}\left(\widehat{\mu}_{t,i}^k > \mu_k + \sqrt{\frac{4\eta\sigma^2 \ln s}{N \sum w_j}}\right)$ is analogous.

$\square$

**Theorem A.3 (Instance Independent Regret Analysis of DDUCB).** *The regret achieved by the DDUCB algorithm is*

$$R(T) \lesssim \sqrt{KTN\sigma^2 \ln(TN)} + K \frac{N\Lambda \ln N}{\sqrt{\ln(1/|\lambda_2|)}},$$

*where $\Lambda$ is an upper bound on the gaps $\Delta_k$, $k = 1, \ldots, K$. Here, $\lesssim$ does not only hide constants but also $\eta$ and $\varepsilon$.*

*Proof.* Define $D_1$ as the set of arms such that their respective gaps are all less than $\sqrt{\frac{K}{TN}\sigma^2 \ln(TN)}$ and $D_2$ as the set of arms that are not in $D_1$. Then we can bound the regret incurred by pulling arms in $D_1$, in the following way

$$\sum_{k \in D_1} \mathbb{E}[n_T^k]\Delta_k \leq \sqrt{\frac{K}{TN}\sigma^2 \ln(TN)} \sum_{k \in D_1} \mathbb{E}[n_T^k]$$

$$\leq \sqrt{KTN\sigma^2 \ln(TN)}$$

Using Theorem 3.2 we can bound the regret obtained by the pulls done to arms in $D_2$:

$$\sum_{k \in D_2} \mathbb{E}[n_T^k] \Delta_k \lesssim \sum_{k \in D_2} \frac{\sigma^2 \ln(TN)}{\Delta_k} + \frac{N \ln(N)}{\sqrt{\ln(1/|\lambda_2|)}} \Delta_k$$

$$\leq \sum_{k \in D_2} \sqrt{\frac{TN\sigma^2 \ln(TN)}{K}} + \frac{N\Lambda \ln(N)}{\sqrt{\ln(1/|\lambda_2|)}}$$

$$\leq \sqrt{KTN\sigma^2 \ln(TN)} + K \frac{N\Lambda \ln(N)}{\sqrt{\ln(1/|\lambda_2|)}}.$$

Adding the two bounds above yields the result.

$\square$

## B   Extended Version of Theorem 3.2

**Theorem 3.2.** *Let $P$ be a communication matrix with real eigenvalues such that $\mathbb{1}^\top P = \mathbb{1}^\top$, $P\mathbb{1} = \mathbb{1}$ whose second largest eigenvalue measured in absolute value is $\lambda_2$, with $|\lambda_2| < 1$. Let $\eta > 1$, and let $\varepsilon > 0$ be arbitrarily small and less than $\frac{\eta-1}{7(\eta+1)}$. Consider the distributed multi-armed bandit problem with $N$ nodes, $K$ actions and subgaussian rewards with variance proxy $\sigma^2$. The algorithm DDUCB with $C = \lceil \frac{\ln(2N/\varepsilon)}{\sqrt{2 \ln|\lambda_2|^{-1}}} \rceil$ and upper confidence bound with exploration parameter $\eta$ satisfies*

1. *The finite-time bound on the regret:*

$$R(T) < \sum_{k:\Delta_k>0} \frac{16\eta(1+2\varepsilon)\sigma^2 \ln(TN)}{\Delta_k} + \left( N(6C+1) + \frac{2\eta}{(\eta-1)} \right) \sum_{k=1}^{K} \Delta_k.$$

2. *The sharper finite-time bound on the regret:*

$$R(T) < \sum_{k:\Delta_k>0} \frac{16\eta(1+2\varepsilon)\sigma^2 \ln(TN)}{\Delta_k}$$

$$+ \left( N(2C+1) + \frac{2NC}{K^\eta} \left( 1 + \frac{1}{N^\eta} \right) + \frac{2\eta}{(\eta-1)(NC)^{\eta-1}} \right) \sum_{k=1}^{K} \Delta_k.$$

3. *The corresponding asymptotic bound:*

$$R(T) \lesssim \sum_{k:\Delta_k>0} \frac{\eta(1+\varepsilon)\sigma^2 \ln(TN)}{\Delta_k} + \left( \frac{N \ln(N/\varepsilon)}{\sqrt{\ln(1/|\lambda_2|)}} + \frac{\eta}{\eta-1} \right) \sum_{k=1}^{K} \Delta_k.$$

4. *In particular, if we choose the value $\eta = 2$ and we choose $\varepsilon = \frac{1}{22}$ we have the finite-time bound*

$$R(T) < \sum_{k:\Delta_k>0} \frac{32(1+1/11)\sigma^2 \ln(TN)}{\Delta_k} + \left( N(6C+1) + 4 \right) \sum_{k=1}^{K} \Delta_k.$$

5. *The corresponding asymptotic bound*

$$R(T) \lesssim \sum_{k:\Delta_k>0} \frac{\sigma^2 \ln(TN)}{\Delta_k} + \frac{N \ln(N)}{\sqrt{\ln(1/|\lambda_2|)}} \sum_{k=1}^{K} \Delta_k.$$

6. *With the same choice of $\eta$ and $\varepsilon$, mixing $\beta_i$ and $b_i$ in an unaccelerated way (cf. Algorithm 4) with $C = \lceil \ln(N/\varepsilon)/\ln(1/|\lambda_2|) \rceil$ the regret is :*

$$R(T) \lesssim \sum_{k:\Delta_k>0} \frac{\sigma^2 \ln(TN)}{\Delta_k} + \frac{N \ln(N)}{\ln(1/|\lambda_2|)} \sum_{k=1}^{K} \Delta_k.$$

The proof given in Appendix A.1 yields all these results straightforwardly. For Theorem 3.2.6, Equation (2) must be used instead of Lemma 3.1. Note that for the unaccelerated version of the algorithm the denominator of the second summand of the regret does not contain a square root. The accelerated version of the algorithm improves on this by making the dependence of the regret on the spectral gap much smaller in the regimes in which the spectral gap is small, that is, when the problem is harder. This is very important for scaling issues, since the spectral gap decreases with $N$ in many typical topologies.

## C  Example for the Regret Comparison in Remark 3.4

If we take $P$ to be symmetric, it is $\sum_{j=1}^{N} \frac{\varepsilon_c^j}{N} = \sum_{j=2}^{N} \frac{\lambda_j^2}{1-\lambda_j^2}$. Consider the graph $G$ to be a cycle with an odd number of nodes (and greater than 1) and take as $P$ the matrix such that $P_{ij} = 1/2$ if $i = j \pm 1$ mod $N$ and $P_{ij} = 0$ otherwise. Then $P$ is a circulant matrix and their eigenvalues are $\cos(2\pi j/N)$, $j = 0, 1, \ldots, N-1$. Then $\frac{\lambda_2^2}{1-\lambda_2^2} = \cot^2\left(\frac{2\pi}{N}\right) \geq \frac{N^2}{4\pi^2} - \frac{2}{3}$ and $\frac{\lambda_3^2}{1-\lambda_3^2} = \cot^2\left(\frac{4\pi}{N}\right) \geq \frac{N^2}{16\pi^2} - \frac{2}{3}$.

As a consequence, $B$ is greater than the corresponding summand in Theorem 3.2.3 in our bound by at least a summand which is $\Theta(N^{7/2})$. On the other hand our summand is $\Theta(N^2 \log N)$. In addition, $A$ is greater than the corresponding summand in Theorem 3.2.3 by a factor of $\Theta(N^2)$.

The bounds above can be proven by a Taylor expansion: $x^2 \cot^2\left(\frac{1}{x}\right) = 1 - \frac{2x^2}{3} + \frac{\xi^4}{15}$, for $x > 0$ and $\xi \in [0, x]$. So $\cot^2\left(\frac{1}{x}\right) \geq \frac{1}{x^2} - \frac{2}{3}$. The bounds above are the latter for $x = \frac{N}{2\pi}$ and $x = \frac{N}{4\pi}$.

## D  Case $|\lambda_2| < 1/e$ in the comparison of the term B in Remark 3.4

He finish here the case left in the comparison performed in Remark 3.4. As in the main paper, note that $\frac{\gamma}{\gamma-1} \geq 1$ and $\frac{1}{1-|\lambda_2|} \geq \frac{1}{\ln(|\lambda_2|^{-1})}$ so we lower bound $B$ for comparison purposes as $B \geq N\left(1 + \frac{\lambda_2'}{\ln(\sqrt{N}/\lambda_2')}\right)$, where $\lambda_2' := \sqrt{N}|\lambda_2| \in [0, \sqrt{N})$. We use the regret in Theorem 3.2.6 for which, regardless of the value of $\lambda_2$, the second summand of the regret multiplying $\sum_{k=1}^{K} \Delta_k$ is $N \ln N / \ln(1/|\lambda_2|) \leq 2B$. The inequality is true since the second line below is holds

$$2B \geq 2N\left(1 + \frac{\lambda_2'}{\ln(\sqrt{N}/\lambda_2')}\right) \geq \frac{N \ln N}{\ln(\sqrt{N}/\lambda_2')}$$
$$\Leftrightarrow \ln N - 2\ln(\lambda_2') + 2\lambda_2' \geq \ln N.$$

Note that if $|\lambda_2| > 1/e$, the accelerated version of the algorithm incurs even lower regret (cf. Theorem 3.2.3). Problems with a lower spectral gap (i.e. greater $|\lambda_2|$) are harder problems since the mixing process is slower. The accelerated version improves on the harder regime $|\lambda_2| \in [1/e, 1)$.

## E  Variants of DDUCB

Here we present the code used to exemplify variants proposed in Remark 3.5. Algorithm 3 ensures further mixing by the property $\|P^s v - \mathbb{1}/N\|_2 \leq |\lambda_2|^s$ explained in Equation (2).

---

**Algorithm 3** Unaccelerated communication and mixing step. unaccel_mix$(x_i, i)$

---
1: Send $x_i$ to neighbors
2: Receive corresponding values $x_j, \forall j \in \mathcal{N}(i)$
3: return $\sum_{j \in \mathcal{N}(i)} P_{ij} x_j$

---

---
**Algorithm 4** Unaccelerated version of Decentralized Delayed UCB at node $i$.
---
1: $\zeta_i \leftarrow (X_i^1, \ldots, X_i^K)$ ; $z_i \leftarrow (1, \ldots, 1)$
2: $C = \lceil \ln(N/\varepsilon)/\ln(1/|\lambda_2|) \rceil$
3: $\alpha_i \leftarrow \zeta_i/N$ ; $a_i \leftarrow z_i/N$ ; $\beta_i \leftarrow \zeta_i$ ; $b_i \leftarrow z_i$
4: $\gamma_i \leftarrow \mathbf{0}$ ; $c_i \leftarrow \mathbf{0}$ ; $\delta_i \leftarrow \mathbf{0}$ ; $d_i \leftarrow \mathbf{0}$
5: $t \leftarrow K$ ; $s \leftarrow K$
6: **while** $t \leq T$ **do**
7: $\quad$ $k^* \leftarrow \arg\max_{k \in \{1,\ldots,K\}} \left\{ \frac{\alpha_i^k}{a_i^k} + \sqrt{\frac{2\eta\sigma^2 \ln s}{N a_i^k}} \right\}$
8: $\quad$ **for** $r$ from 0 to $C - 1$ **do**
9: $\quad\quad$ $u \leftarrow$ Play arm $k^*$, return reward
10: $\quad\quad$ $\gamma_i^{k^*} \leftarrow \gamma_i^{k^*} + u$ ; $c_i^{k^*} \leftarrow c_i^{k^*} + 1$
11: $\quad\quad$ $\beta_i \leftarrow$ unaccel_mix$(\beta_i, r, i)$ ; $b_i \leftarrow$ unaccel_mix$(b_i, r, i)$
12: $\quad\quad$ $t \leftarrow t + 1$
13: $\quad\quad$ **if** $t > T$ **then**
14: $\quad\quad\quad$ return
15: $\quad\quad$ **end if**
16: $\quad$ **end for**
17: $\quad$ $s \leftarrow (t - C)N$
18: $\quad$ $\delta_i \leftarrow \delta_i + \beta_i$ ; $d_i \leftarrow d_i + b_i$ ; $\alpha_i \leftarrow \delta_i$ ; $a_i \leftarrow d_i$
19: $\quad$ $\beta_i \leftarrow \gamma_i$ ; $b_i \leftarrow c_i$ ; $\gamma_i \leftarrow \mathbf{0}$ ; $c_i \leftarrow \mathbf{0}$
20: **end while**
---

---
**Algorithm 5** Accelerated Decentralized Delayed UCB at node $i$ with some variants.
---
1: $\zeta_i \leftarrow (X_i^1, \ldots, X_i^K)$ ; $z_i \leftarrow (1, \ldots, 1)$
2: $C = \lceil \ln(2N/\varepsilon)/\sqrt{2\ln(1/|\lambda_2|)} \rceil$
3: $\alpha_i \leftarrow \zeta_i/N$ ; $a_i \leftarrow z_i/N$ ; $\beta_i \leftarrow \zeta_i$ ; $b_i \leftarrow z_i$
4: $\gamma_i \leftarrow \mathbf{0}$ ; $c_i \leftarrow \mathbf{0}$ ; $\delta_i \leftarrow \mathbf{0}$ ; $d_i \leftarrow \mathbf{0}$
5: $t \leftarrow K$ ; $s \leftarrow K$
6: **while** $t \leq T$ **do**
7: $\quad$ **for** $r$ from 0 to $C - 1$ **do**
8: $\quad\quad$ $k^* \leftarrow \arg\max_{k \in \{1,\ldots,K\}} \left\{ \frac{\alpha_i^k}{a_i^k} + \sqrt{\frac{2\eta\sigma^2 \ln s}{N a_i^k}} \right\}$
9: $\quad\quad$ $u \leftarrow$ Play arm $k^*$, return reward
10: $\quad\quad$ $\gamma_i^{k^*} \leftarrow \gamma_i^{k^*} + u$ ; $c_i^{k^*} \leftarrow c_i^{k^*} + 1$
11: $\quad\quad$ $\beta_i \leftarrow$ mix$(\beta_i, r, i)$ ; $b_i \leftarrow$ mix$(b_i, r, i)$
12: $\quad\quad$ $t \leftarrow t + 1$
13: $\quad\quad$ // It also works adding this:
14: $\quad\quad$ // $\alpha_i^{k^*} \leftarrow \alpha_i^{k^*} + u/N$ ; $a_i^{k^*} \leftarrow a_i^{k^*} + 1/N$
15: $\quad\quad$ // $s \leftarrow s + 1$
16: $\quad\quad$ // and / or this:
17: $\quad\quad$ // $\delta_i \leftarrow$ unaccel_mix$(\delta_i, i)$ ; $d_i \leftarrow$ unaccel_mix$(d_i, i)$
18: $\quad\quad$ **if** $t > T$ **then**
19: $\quad\quad\quad$ return
20: $\quad\quad$ **end if**
21: $\quad$ **end for**
22: $\quad$ $s \leftarrow (t - C)N$
23: $\quad$ $\delta_i \leftarrow \delta_i + \beta_i$ ; $d_i \leftarrow d_i + b_i$ ; $\alpha_i \leftarrow \delta_i$ ; $a_i \leftarrow d_i$
24: $\quad$ $\beta_i \leftarrow \gamma_i$ ; $b_i \leftarrow c_i$ ; $\gamma_i \leftarrow \mathbf{0}$ ; $c_i \leftarrow \mathbf{0}$
25: **end while**
---

# F Estimation of the number of nodes

The total number of nodes can be estimated at the beginning of the algorithm, with high probability. Given a value per node, the gossip protocol allows for the computation at each node of the average of those values. If a node starts with a number $u \neq 0$ and the rest of the nodes start with the value 0, then using the gossip protocol after some iterations makes the nodes hold an approximation of the value $u/N$. The approximation improves exponentially in the number of steps (cf. Equation (2), for instance) and does not depend on $N$, but in the spectral gap. To recover $N$, the value $u$ is broadcast at the same time the values are being mixed, so at each time step a node receives from her neighbors the mixing value and $u$. Since we want to run this procedure in a decentralized way, we cannot tell which node should start with the value $u$, so we make every node compute a number $u_i$ at random and they start broadcasting and mixing it. However, during the mixing process, we make each node only keep and mix the value corresponding to the minimum $u_i$ so at the end of this process, each node only contains $u = \min u_i$ and the approximate value of $u/N$, if no two nodes started with $\min u_i$, which only occurs with low probability. The procedure can be repeated to increase the probability of success.

The approximations of $N$ can be broadcast and nodes could use the minimum and maximum as lower and upper bounds on $N$. The algorithm really only needs upper and lower bounds on $N$. The delay constant $C$ would be computed with the upper bound on $N$ and the upper confidence bound would be computed using the lower bound, which translates to using a greater exploration parameter $\eta$. Since our analysis was done in general for the delay and the exploration parameters, the bounds in Theorem 3.2 hold, substituting the delay and exploration parameters by the new values.

# G Experiments

We have observed that the accelerated method performs well with the recommended values for the exploration parameter $\eta$ and the parameter $\varepsilon$ that measures the precision of the mixing after a stage. So we use these recommended values, that are $\eta = 2, \varepsilon = \frac{1}{22}$.

The distributions of the arms in the bandit problem used in the experiments are Gaussian with variance 1. There is one arm with mean 1 and 16 other arms with mean 0.8. We have executed the algorithms for cycle graphs of size 100 and 200 and for square grids of size 100 and 225. We compare executions of coopUCB [25] using different exploration parameters with $DDUCB$ with the fixed exploration parameter $\eta = 2$ and observe that DDUCB outperforms every execution of coopUCB. Each algorithm for each setting was executed 10 times. Average regret is shown in the figures. The experiments we present are representative of the regret behavior we have observed in a greater variety of scenarios upon different choices of means, number of arms and variance. Using different exploration parameters for coopUCB did not make it show a behavior as effective as the one observed for DDUCB. We tuned $\gamma$, the exploration parameter of coopUCB, to get best results for that algorithm and report also $\gamma = 2$ for comparison purposes. In the figures one can observe that after a few stages, DDUCB algorithms learn with high precision which the best arm is and the regret curve that is observed afterwards shows an almost horizontal behavior. After 10000 iterations, coopUCB not only accumulates a greater regret but the slope indicates that it still has not learned effectively which the best arm is. The graphs for coopUCB $\gamma = 1.01$ and $\gamma = 1.001$ are very similar but the smaller $\gamma = 1.001$ seems to work slightly better.

The constant $C$ for the unaccelerated method is usually excessively large, so we found it is convenient to heuristically decrease it, or equivalently to use a different value of $\varepsilon$. Experiments shown below, Figure 2 and Figure 3, are the same as the plots in the main paper in a bigger format. The parameter $\varepsilon$ for the unaccelerated method was picked so that the value of $C$ for is the same for both the unaccelerated and the accelerated method. We used the recommended modification of DDUCB consisting of adding to the variables $\alpha_i$ and $a_i$ the information of the pulls that are done times $1/N$

The matrix $P$ was chosen according to [14]. That is, we define the graph Laplacian as $\mathcal{L} = I - D^{-1/2}AD^{-1/2}$, where $A$ is the adjacency matrix of the communication graph $G$ and $D$ is a diagonal matrix such that $D_{ii}$ contains the degree of node $i$. Then for regular graphs, if we call $\delta$ the common degree of every node, we pick $P = I - \frac{\delta}{\delta+1}\mathcal{L}$. For non regular graphs, like the square grid we used, letting $\delta_{\max}$ be the maximum degree of the nodes we pick $P = I - \frac{1}{\delta_{\max}+1}D^{1/2}\mathcal{L}D^{1/2}$. These matrices always satisfy the assumptions needed on $P$. For reference, for our choice of $P$, the

inverse of the spectral graph of the cycle is $O(N^2)$ and it is $O(N)$ for the grid [14]. Note that for an expander graph it is $O(1)$.

Figure 2: Simulation of DDUCB and coopUCB for a cycle graph of 100 nodes (left), 200 nodes (right).

Figure 3: Simulation of DDUCB and coopUCB for a square grid of 100 nodes (left) and 225 nodes (right)

# H   Notation

| | |
|---|---|
| $N$ | Number of agents. |
| $T$ | Number of time steps. |
| $K$ | Number of actions. |
| $P$ | Communication matrix. |
| $\lambda_1, \lambda_2, \ldots, \lambda_N$ | Eigenvalues of $P$ sorted by norm, i.e. $\lvert\lambda_1\rvert > \lvert\lambda_2\rvert \geq \lvert\lambda_3\rvert \geq \cdots \geq \lvert\lambda_n\rvert$. It is always $\lambda_1 = 1 > \lvert\lambda_2\rvert$. |
| $\mu_1 \geq \mu_2 \geq \cdots \geq \mu_K$ | Means of arms' distributions. |
| $\Delta_i$ | Reward gaps, i.e. $\mu_1 - \mu_i$. |
| $\alpha_i, a_i$ | Normalized delayed sum of rewards and number of pulls that are mixed. Normalization accounts for a division by $N$. |
| $\beta_i, b_i$ | Sum of rewards and number of pulls done that are being mixed. |
| $\gamma_i, c_i$ | Sum of new rewards and new number of pulls. |
| $\pi_t^k$ | Vectors in $\mathbb{R}^N$ whose $i$-$th$ entry is the current reward obtained at time $t$ by node $i$ if arm $k$ was pulled; it is 0 otherwise. |
| $p_t^k$ | Vector in $\mathbb{R}^N$ whose $i$-$th$ entry is 1 if arm $k$ was pulled at time $t$ by node $i$, or 0 otherwise. |
| $C$ | Number of steps that define the stages of the algorithm. |
| $X_j^k$ | Reward obtained by pulling arm $k$ for the $j$-$th$ time. If arm $k$ is played several times in one time step lower indices are assigned to agents with lower indices. |
| $n_{t,i}^k$ | Number of times arm $k$ is pulled by node $i$ up to time $t$. |
| $n_t^k$ | $\sum_{i=1}^{n} n_{t,i}^k$. |
| $m_t^k$ | Sum of rewards coming from all the pulls done to arm $k$ by the entire network up to time $t$. |
| $I_{t,i}$ | Action played by agent $i$ at time $t$. |
| $J_t^k$ | Number of rewards that come from arm $k$ that were computed up to $C$ steps before the last time the variables $\alpha_i$ and $a_i$ were updated if current time is $t$. |