[Reviews · NeurIPS 2019]

Reviewer 1



The authors propose a decentralized algorithm for stochastic bandits. The key idea is to use information that is mixed to build the upper confidence interval of each arm. The regret bound of their algorithm equals the optimal regret for N agents plus a term depending on the spectral gap of the communication graph, and the communication cost is O(K) per round for each agent. Besides, their empirical results also show the good performance of their algorithm. I am concerned about a tighter lower bound relevant to the graph structure. I mean, the graph structure may cause communication latency, which results in a higher regret. However, the lower bound proposed in the paper doesn't indicate the dependence on graph structure, which is derived by comparing with the centralized UCB algorithm for TN steps. I believe that a tighter lower bound relevant to graph structure can indicate whether the regret can be further improved. I also wonder whether the communication cost can be improved. In DDUCB, each agent needs to communicate with other agents every round. As a result, the communication cost is linear in T. Is it possible to communicate only in certain rounds without loss of the regret? For example, for a single-agent stochastic MAB problem, we can only update the mean rewards and counter of an arm when the number of pulling that arm is doubled. This will not cause much loss of the regret. I don't know whether this idea can be used to reduce the communication complexity. Overall, I believe this paper is interesting and well-organized, and I vote for accepting it.

Reviewer 2



This paper studies the decentralized stochastic bandits, where N players can pull any arm of their choice at each time step. They then receive a corresponding reward (two players pulling the same arm can receive two different rewards in this setting) and can communicate after each time step through a given (possibly unknown) communication graph. This problem has already been studied, especially by Landgren et al., who used a gossip based algorithm. The authors here propose an improved gossip algorithm which yields a lower regret while requiring less prior knowledge of the graph. The paper uses interesting techniques such as gossip acceleration. Unfortunately, this work presents several drawbacks: - The results seem incremental as the used techniques are similar to those of Landgren et al. - Despite the authors claims, I am not convinced that this new algorithm brings a significant regret improvement (explained below). - I have a general feeling that some parts of the paper are unnecessary complex. They bring a lot of confusion without being significant for the results (or while they can be rewritten in a simpler way). For these different reasons, I think that this paper can still be improved. ------------------------------------------------------ Major comments: 1. Theorem 3.2.6 (given in appendix) gives the same asymptotic upper bound for the unaccelerated version of DDUCB. Then, what is the significance of acceleration in DDUCB ? From the proof, it seems that it will only improve the regret by some constant factor. This point is the main example of unnecessary complexity I mentioned above. If the acceleration only improves the regret by some constant, I think it is not worth the complexity it adds to the paper, and it could be delayed to the appendix (and bring the simpler unaccelerated version to the main text). All of this is supported by the experiments, where the unaccelerated version is comparable to the accelerated one. 2. What I understand from the paragraph at the beginning of page 5 is that the novelties of this paper compared to the work of Landgren et al. are the following: - the acceleration technique (which is not so useful if theorem 3.2.6 is correct) - the observations are not directly mixed to the statistics but we wait some time instead - the mathemetical tools for the proof of the upper bound are different This is why I consider this work incremental, especially if the acceleration is not so useful. 3. My other point is that I am still not convinced of the significance of improvement compared to the previous work. This point is discussed in Remark 3.4: the authors claim they significantly improve the incurred regret. But when looking at the details of their claim (see Appendix C), they show it by considering that in a cycle graph, the communication matrix P will be 1/2 above and below the diagonal and 0 elsewhere. But why would we consider such a matrix P ? Intuitively, the matrix with 1/3 below, above and on the diagonal will give better communication. After looking at "On Distributed Cooperative Decision-Making in Multiarmed Bandits" , it indeed seems that they will choose 1 - kappa on the diagonal and kappa/2 above and below for some kappa in (0, 1]. They do not seem to precise which kappa to choose then, but I do not believe that 1 is then optimal in the cycle graph case. Also, which P is used in the experiments ? With the given parameters and the P you mention for cycle graphs, I believe the difference in regret should be larger between coop-UCB and DDUCB for the cycle graphs. Furthermore, the two values for N (200 and 225) are too close to really observe the behavior with N. ------------------------------------------------------------------ Minor comments 4. Page 3 line 133. You say that the size of each message could be more than poly(K). This is true for an arbitrary d but you then claim that d is actually set as \sqrt{K}, so it is in poly(K). Furthermore, you forgot an important additional complexity in the cited paper: there are delayed feedbacks. 5. page 4 line 155. Do you have a practical justification of why you do not want to build global structures ? To me, this would be the optimal way to build a decentralized algorithm in your setting, so a solid justification of why we do not want it (in practice) would be welcome. 6. page 4 line 163: this is not really an interval. This is actually the upper bound of the confidence interval. 7. Page 4: don't we need P to have positive coefficients as well ? 8. Page 5 line 194: this is for v in the simplex. I don't think m/N and n/N are in the simplex. The link between the two points is not clear here. 9. typo Page 5 line 195 : "run 2K running consensus" 10. For algorithm 2, I think it would be easier to describe it using directly the coefficients of q_C instead of the w here. We here need to read the appendix to know the link between q_C and w, so I think it would be better to directly put q_C here and refer to the appendix for its coefficients. This unnecessarily complexifies the algorithm to me. 11. Also, the index i seems useless for w here (and in the appendix, you add an index k to w, which also seems useless). 12. Page 7 line 285: why is there a +1 ? Here it is unclear whether this is a heuristic to give some intuition of what the lower bound should be or a proof of what the lower bound is. 13. Page 8: I find the paragraph "Variants of DDUCB" very dense and unclear for some parts and this part does not really enter in the structure of the paper. I think it can be shortened/delayed to the appendix. The details could then be made clearer in the appendix, and you would save some space in the main text. 14. Page 1 supp: is the s_t line 482 the same as line 486 ? I believe yes but this is unclear 15. I do not understand what is shown in Section D of the appendix. This section is very confusing, and needs to be rewritten in a clearer way. 16. It seems there are format errors in the pseudocodes of the appendix 17. How did you choose the parameter gamma of coop-UCB for your experiments ? 18. I think it could be good for the reader to add (in the appendix) a table, giving examples of the used communication matrix P and the corresponding lambda_2 for some classical graphs. ------------------- In their answer, the authors pointed out the regret improvement brought by the acceleration method and thus answered to my major concern. They also answered to some of my other concerns (no global structure, the regret is significantly better than Landgren et al.) and I thus decide to raise my score in consequence. I hope the authors will take into account all the reviewer comments for the camera ready version (in case of acceptance).

Reviewer 3



-- In general, distributed algorithms are measured in terms of the size of messages or by the number of messages. This is not the case here: the only measure taken into account is the regret. I do no understand why the nodes then just not send a vector of O($K$) reals that corresponds to some estimation of the reward of arms. Why can every agent not broadcast the following information: for every agent i (if he knows it), for each arm a , the number of times arm a has been selected by agent i, plus estimation of the reward for a. If these informations are available (and of course updated) why matrix P is required? One must take into accounts the cost of communications. Without these taken into considerations, this is very hard to compare the various algorithms. -- the paragraph that starts at line 227 (page 5) should be rewritten. -- How matrix P is computed? Is that expected to be some prior knowledge of the network, or should it be computed in distributed manner? If so, it would be good to consider also the related communication costs (evaluate the quantity of informations that will transit). -- Page 4, line 190 ''See [37] for a proof and for a discussion on how to choose matrix P ''. Seeing the role played by the matrix P in this article, the intuition about its meaning should be explained in the current article, otherwise the current article is of very weak interest. I hence disagree that the current paper refers to other articles about how it is built and chosen. -- The work related to experimentation is very limited (even with the details in appendix). There is no explanation why the algorithm has good performances compared to other algorithms. My opinion is that the topologies that are used are graphs whose nodes have mainly a same type of neighborhood. It would have been good to understand the performance with respect to the ratio ''number of vertices/diameter`''. Other comments/questions just listed: --- In the experimental part, the topologies are very regular graphs and that means that matrix $P$ is very regular (an probably too regular). Would it be possible (if the paper is accepted) to do some experimentations on less regular graphs. --- Are the agents supposed to have distinct identifiers? If this is not the case, I do not understand how the algorithm is working. There exist some impossibility results: for example : Julien M. Hendrickx, John N. Tsitsiklis: Fundamental limitations for anonymous distributed systems with broadcast communications. Allerton 2015: 9-16

[Author Response · NeurIPS 2019]

**Reviewer 1.** We agree with **R1** that the two directions mentioned are natural. **(1)** Our **lower bound** is likely to be too loose. Note however that the factor in the second summand of regret for no communication (cf. line 278) is $N$, *irrespective of the graph structure*. A question would be what the second summand can be if the first one is proportional to $\log(TN)$. We have not found yet a way to translate the impossibility of mixing information arbitrarily quickly to a lower bound on the regret. It is tricky given the statistical properties of our problem, in which as time grows, neither mixing nor the graph should be that important. At the same time, establishing *graph-dependent lower bounds for decentralized methods is an active area of research in its own right*. Despite the vast literature on decentralized convex optimization, for instance, the first graph-dependent lower bounds appeared only recently in the NeurIPS 2018 paper [30]. **(2)** One would also expect that as time grows *less communication should work well enough*. We have explored this path (counting the number of mixing stages that would be needed) and have found it is not straightforward, although we know that this quantity at least has to be linear in $N$ (stages). What plays an important role is that most ways that increase delay (equivalently, reduce communication) introduce a multiplicative factor depending on $N$ to the $\log(TN)$ regret term, which is inadmissible, since to match the same regret $T$ would need an exponential increase. This is because more delay is similar to having a different $\varepsilon$ (line 571).

**Reviewer 2. New techniques, new (not incremental) results and importance of acceleration.** *Our work focuses on the dependence of the regret as a function of the network size, N, not just as a function of the time horizon, T.* In this respect, **our results on the regret are the first of their kind and are not incremental**: **(1)** We have the first decentralized algorithm with a $\log(TN)$ regret term with no factors dependent of the network size and topology. The factor is important because in a comparison, our $T$ would need to be exponentially larger to yield regret as large as the ones where a poly $N$ factor appears in front of the $\log T$ term, as in Landgren et al. **(2) Accelerated method does not yield same regret**. Acceleration gives a square root improvement in the denominator of the second summand, which improves regret w.r.t. $N$ (cf. lines $574-576$). **(3)** Take the example in Appendix C, for instance: when evaluated it yields a factor in the second summand of regret of $\Theta(N^3 \log N)$ (unacc.) instead of $\Theta(N^2 \log N)$ (acc.), which is compared to the worse rate $\Theta(N^{7/2})$ in Landgren et al. This makes a **crucial difference in large networks**. **(4)** About this example, note that the matrix **R2** proposes is only making the process lazy and thus the mixing strictly slower. The new matrix amounts to $\frac{1}{3}I + \frac{2}{3}P$, and the same computations in our paper yield $\lambda_i = \frac{1}{3} + \frac{2}{3}\cos(2(i-1)\pi/N)$, $\frac{\lambda_i^2}{1-\lambda_i^2} \geq \frac{3N^2}{8(i-1)^2\pi^2} - \frac{5}{8}$, which is of the same order as the matrix $P$ we consider, so the exact same asymptotics apply. **(5)** One cannot compare directly the performance of the acc. and unacc. methods in the experiments, since the delay in the unacc. one was picked to be the delay predicted for the *acc. theorem*, as mentioned in the section and indicated in the plot. The acc. one would probably improve if $\varepsilon$ is tuned too. Our theory predicts a gap between the unacc. and acc. methods that does not depend on $T$ and that increases when $N$ grows (i.e. $\lambda_2$ increases). This gap is not constant (it depends on $N$) and it can be seen in the experiments. **(6)** A lot of effort has been put in finding optimal algorithms for distributed systems, and **acceleration techniques are key to achieve optimality**. See for instance [30], which won one of the best paper awards at NeurIPS 2018, and that could find optimality in their setting under local regularity via a Chebyshev acceleration argument as well. See **R3**'s section on global structures.

**Other differences compared to Landgren et al.:** **(7)** We use delays, which give **lower variance estimators**. **(8)** We **use less information**, which is of interest for **decentralization purposes** and has important **computational implications**. As we write in the paper, the algorithm of Landgren et al. requires the full set of eigenvalues and eigenvectors of $P$ to compute $\varepsilon_c$. This requires full knowledge of $P$ and $O(N^3)$ operations. In our case, agents do not need to know the entire matrix $P$ (only the spectral gap of $P$, as typical in decentralized methods [30]) and our comp. cost is constant per iteration and node. Just computing $\varepsilon_c$ in Landgren et al. for $N$ larger than 200 takes several hours. The two algorithms are very different. *We believe our algorithm gives lots of new insights for the understanding of decentralized bandits.*

**Reviewer 3.** The paper focuses on the statistical problem of minimizing regret, something of interest in its own right. Analyzing communication cost is beyond the scope of this paper (see response to **R1.(2)**). The assumptions of the distributed system are included in Sec 2 and they correspond to the standard ones typically considered in the literature on decentralized methods, cf. [30, 31]. Our main focus was theoretical work and we added the experiments as proof of concept rather than an aiming at an exhaustive study. We provided theorems and compared the regrets obtained, the experiments show that the theoretical results carry on in practice. The matrix $P$ is a classical object in decentralized methods. Only the maximum degree of the graph is needed to construct $P$ [31]. We will add details in the final submission. There are no identifiers. In the introduction of the paper that R3 cited one can read that averages are feasible, that is the only thing we use. The comparison of various algorithms is fair, all of them use the same communication. **Decentralization (without global structures and broadcasting) is important**, as reflected by the vast and growing literature in machine learning. **(1)** It is a first step to address time varying graphs or faulty networks (cf. 61), and in systems with privacy/communication constraints. **(2)** It has many applications, like sensor networks (cf. 71 & [31]). The topology is grid-like, agents can only interact with close neighbors, and it is very different from complete graphs where broadcasting can be implemented. We will add a comment on this.

[Meta-Review · NeurIPS 2019]

The paper proposes new algorithmic and theoretical solutions to achieving fast distributed bandit optimization over agents connected over a network, when only agents connected by a network link can exchange messages. The reviewers largely agree that the paper's exposition and results are significantly novel compared to past work on the subject, and the results improved substantially over previous ones as shown for example networks. A concern regarding the impact of acceleration of the gossip subroutine on the derived performance bounds was addressed satisfactorily by the author response. On the other hand, I must remark that a separate concern about connection to the real world -- how to capture/measure in a better and more finer way the communication cost across the entire network -- still remains unaddressed in this line of work thus far. I would urge the author(s) to take a closer look at the issue of accounting for communication at a more granular level other than just allowing real-valued messages across links (perhaps by drawing upon ideas from the communication networks/information theory communities), and expose a more concrete dependence of performance on the information bottlenecks in messaging between adjacent nodes.